# A Review on Electrohydrodynamic (EHD) Pump

**DOI:** 10.3390/mi14020321

**Published:** 2023-01-26

**Authors:** Yanhong Peng, Dongze Li, Xiaoyan Yang, Zisu Ma, Zebing Mao

**Affiliations:** 1Department of Information and Communication Engineering, Graduate School of Engineering, Nagoya University, Furo-cho, Chikusa-ku, Nagoya 464-8603, Japan; 2Department of Intelligent Science and Technology, College of Computer Science and Technology, Qingdao University, 308 Ning Xia Lu, Laoshan District, Qingdao 266071, China; 3School of Computer Science, The University of Sydney, Sydney, NSW 2006, Australia; 4Department of Mechanical Engineering, Tokyo Institute of Technology, 2-12-1 Ookayama Meguro-Ku, Tokyo 152-8550, Japan

**Keywords:** functional fluidic pumps, electrohydrodynamic pumps, flow delivery, robotics

## Abstract

In recent years, functional fluidic and gas electrohydrodynamic (EHD) pumps have received considerable attention due to their remarkable features, such as simple structure, quiet operation, and energy-efficient utilization. EHD pumps can be applied in various industrial applications, including flow transfer, thermal management, and actuator drive. In this paper, the authors reviewed the literature surrounding functional fluidic and gas EHD pumps regarding the following aspects: the initial observation of the EHD effect, mathematical modeling, and the choice of pump structure, electrode configuration, and working medium. Based on the review, we present a summary of the development and latest research on EHD pumps. This paper provides a critical analysis of the current limitations of EHD pumps and identifies potential areas for future research. Additionally, the potential application of artificial intelligence in the field of EHD pumps is discussed in the context of its cross-disciplinary nature. Many reviews on EHD pumps focus on rigid pumps, and the contribution of this review is to summarize and analyze soft EHD pumps that have received less attention, thus reducing the knowledge gap.

## 1. Introduction

Pumps have proven to be a vital component in various micromotors and actuators over the last several decades, and promoted the development of various mechanical equipment. The EHD effect directly converts electrical energy into kinetic energy. EHD pumps bearing functional fluid or gas that operate based on the EHD effect have received much attention recently. However, there are few papers reviewing the recent literature on soft EHD pumps with the rapid rise of this field. This paper reviews the mechanism, structures, fluid types, and current progress of functional fluidic EHD pumps and EHD gas pumps, including soft pumps, as well as prospects for future development. We provide a reference for the further development of this emerging field by outlining relatively mature EHD-based research. Our contribution is to summarize and analyze soft and hard EHD pumps and help reduce current knowledge gaps.

## 2. Brief History of the EHD Pump

EHD effects were first discovered in 1629 by Cabeo, who observed that electric fields could attract sawdust, after noticing repulsion following physical contact [1]. However, Cabeo did not understand the rationale behind this phenomenon. Over the next few hundred years, Wilson investigated the EHD effect, coining the term electric wind, and attempted to find practical applications [2]. In 1961, a theory relating to ionic wind was developed by Robinson, who proposed the possibility of developing functional electrostatic blowers, namely EHD gas pumps [3]. Although this electrostatic blower had the advantages of low wear, quiet operation, and no requirement for lubrication as compared to conventional fans (i.e., a mechanical pump), Robinson was disappointed by the inability to overcome the extremely low operating efficiency (the conversion efficiency was less than 1% from corona discharge to mechanical energy). The low conversion efficiency from electrical to mechanical energy represented a primary limitation of EHD pumps. Many researchers attempted to improve the device structure using EHD pumps, and there have been some experimental studies focused on improving the flow rate and efficiency of EHD pumps. In 2008, Moreau et al. made substantial improvements [4]. Morrow et al. increased the electrical energy conversion efficiency to 1.72% via the pin-ring and pin-grid electrode configuration, which provided more than twice the conversion efficiency compared to previous EHD effect studies. In recent times, the higher energy conversion efficiency of EHD pumps has also been demonstrated by June et al. [5]. June et al. experimentally tested a five-needle mesh-like EHD pump prototype and found static efficiencies of up to 14%.

In addition to EHD air pumps, functional fluidic EHD pumps have also been developed in recent decades. Sharbaugh et al. [6] were the first to use media other than air in EHD pumps, attempting to apply EHD pumps to transformers. However, this was deemed impractical owing to subsequent contamination of the collector electrode after hundreds of hours of operation. Five-years later, Crowley et al. were the first researchers to publish an experimental study on the behavior of multiple working fluids using a rectangular EHD pump with a parallel strip-to-strip electrode [7]. This paper develops a basic EHD pumping model to illustrate the working fluid’s material properties. They applied the model to several experimental pumps reported earlier and found good agreement. Examples of this model demonstrate several new fluids suitable for EHD pumping. The authors concluded that there are considerable differences in the maximum velocity that can be achieved for different fluids and that the maximum velocity for a given fluid cannot determine the maximum efficiency. The basic geometry of a functional fluidic EHD pump is depicted in Figure 1. Several years later, a multi-stage EHD pump using transformer oil as the working fluid was designed as a practical solution for motor cooling [8,9]. There are also studies using a dielectric fluid (dibutyl sebacate, an organic plasticizer) as a medium, both as a single-stage and a multi-stage configuration of the same geometry. Researchers in other studies also employed a dielectric fluid (such as dibutyl sebacate) as a medium, both as single- and multi-stage configurations of the same geometry [10,11].

Despite the computational conditions, the flawed theories of physics and materials science, and the limitations of experimental equipment, previous explorations built the foundation for future research, as these results retain the majority of their validity even today.

## 3. Principles

### 3.1. Fluid Theory and Chemical Synthesis Methods

There is a reversible process of dissociation and recombination between the neutral substances of this structure and their corresponding positive and negative ions. These generated positive and negative ions act as charge carriers between parallel electrodes. For example, a neutral species AB, and its positive $A^{+}$ and negative $B^{-}$ ions have the following relation: (1)$$AB\overset{\begin{array}{c} Disocciation\\ Recombination \end{array}}{⟷}A^{+}+B^{-}$$

The dissociation–recombination process of the electrolyte produces the only source of ions in the liquid, and the electric field in the EHD conduction pump is not strong enough to generate charge injection from the electrodes [12]. This model consists of reversible processes of dissociation and recombination of neutral species that can be expressed as a state in equilibrium [13]: (2)$$k_{D}c_{0}=k_{R}n_{+}^{eq}n_{-}^{eq}=k_{R}{\left(n_{eq}\right)}^{2}$$
where $c_{0}$ is the concentration of neutral species and $n_{+}^{eq}$ and $n_{-}^{eq}$ are the concentrations of positive and negative species (in equilibrium, $n_{+}^{eq}=n_{-}^{eq}=n_{eq}$). The dielectric liquid of the EHD conduction pump is non-polar or weakly polar, and the concentration $c_{0}$ of neutral substances can be considered constant. $k_{D}$ and $k_{R}$ are the dissociation and recombination rates, respectively.

The dissociation rate increases when an external electric field |E| is applied, owing to the Onsager–Wien effect [14]: (3)$$k_{D}\left(|E|\right)=k_{D}^{0}F(w(|E|))$$
where *F* is the Onsager function (Equation 4) and w(|E|) is the enhanced dissociation rate coefficient (Equation 5),
(4)$$F(w(|E|))=\frac{I_{1}\left(4w(|E|)\right)}{2w(|E|)}$$
(5)$$w\left(|E|\right)=\frac{L_{B}}{L_{O}}={\left(\frac{e_{0}^{3}\left|E\right|}{16\pi \varepsilon k_{B}^{2}T^{2}}\right)}^{\frac{1}{2}}$$

$I_{1}\left(x\right)$ is the first kind modified Bessel function with order 1. $L_{B}$ and $L_{O}$ are the Bjerrum and Onsager distances, respectively, [15],
(6)$$L_{B}=\frac{e_{0}^{2}}{8\pi \varepsilon k_{B}T},\;L_{O}=\sqrt{\frac{e_{0}}{4\pi \varepsilon |E|}}$$

$L_{B}$ is the distance between two ions, where the electrostatic energy becomes of the same order as the thermal energy. When two ions are separated by less than $L_{B}$, they can be considered bounded in an ion pair sine thermal motion cannot overcome their electrical attraction. The length $L_{O}$ is the distance from the point charge at which the magnitude of the external electric field |E| is of the same order as the electric field produced by the charge in the working medium. In (Equation 6), ε is the absolute permittivity (equal to the product of the permittivity of liquid and that of vacuum). For an external electric field to affect the equilibrium of an associated ion pair, the external electric field must be strong enough to affect the electrical attraction between ions in the associated ion pair ($L_{O}\leq L_{B}$). When |E|=0, we have w(0)=0 and F(0)=1, and no enhancement occurs. In the presence of enhancement, assuming that the dissociation–recombination equilibrium does not change significantly, the equilibrium concentration of the ionic species is [16]:(7)$$n_{eq}=n_{eq}^{0}\sqrt{F(w(|E|)}$$
where $n_{eq}^{0}$ is the concentration of the ionic species without an applied external electric field. Conductivity is proportional to the concentration of ionic species at equilibrium. For symmetric electrolytes, it is
(8)$$\sigma =2e_{0}Kn_{eq}$$
where *K* is the ionic mobility. Moreover, the conductivity with the electric field can be expressed as
(9)$$\sigma =\sigma_{0}\sqrt{F(w(|E|)}$$
where $\sigma_{0}$ is the conductivity without field-enhanced dissociation. According to Mao et al. [17], the selection of working fluid follows the Figure 2. Typically, the dielectric fluid used in EHD pumps has a conductivity between $10^{-11}$ and $10^{-7}$ S/m [12].

### 3.2. Physical Model

The basic EHD pump structure can be represented by two parallel electrodes immersed in a pure dielectric liquid [18], as depicted in Figure 3. The resultant force driving the dielectric medium in the electric field is composed of Coulomb force, dielectric force and electrostrictive force.

Corona discharge in fluids is a complex physical phenomenon that remains yet to be fully understood. However, we can simplify the mathematical problem by setting proper boundary conditions [19]. Combined Poisson’s Equation (Equation 10) with Gauss’ law (Equation 11), the electric field strength is described as: (10)$${▽}^{2}V=-\frac{\rho_{e}}{\varepsilon_{0}}$$
(11)E=−▽V
where $\rho_{e}$, $\varepsilon_{0}$ and *V* are the space charge density, the dielectric permittivity of the fluid and voltage potential of the emitter electrode, respectively. Here, the space charge density $\rho_{e}$ can be expressed by the volume concentration of positive $n_{+}$ and negative $n_{-}$ species: (12)$$\rho_{e}=e_{0}\left(n_{+}-n_{-}\right)$$

The current density j can be described by the following equation because of the current continuity condition: (13)▽·j=0

Then, the current density j passing through the dielectric liquid can be calculated as following:(14)$$j=(\mu_{\varepsilon }\rho_{e}E+\rho_{e}U-D▽\rho_{e})$$
where $\mu_{\varepsilon }$ is the ion mobility, D is the ion diffusion coefficient, and **U** is the velocity of the fluid flow. Mathematical EHD problems can be simplified when proper boundary conditions are set. We can describe the motion of the fluid by the Navier–Stokes Equation [20]: (15)$$\rho \cdot \left[\left(\frac{\partial U}{\partial t}\right)+\left(U\cdot ▽\right)U\right]=-▽p+\mu {▽}^{2}U-\rho ▽V$$
where ρ is the mass density of the fluid, *p* is the pressure, and μ is the dynamic viscosity of the fluid. If the fluid flow has constant density and cannot be compressed, the continuity equation can be expressed as: (16)(▽·U)=0

In addition, the functional fluidic EHD phenomenon is due to the interaction of electric and flow fields in a dielectric fluid medium. The interaction between the electric field and the flow field causes flow motion through a ponderomotive force, and the density of this force can be expressed as follows [21,22]: (17)$$f_{e}=\rho_{e}E-\frac{1}{2}E^{2}▽\varepsilon +\frac{1}{2}▽\left[E^{2}\frac{\partial \varepsilon }{\partial \rho }\rho \right]$$

The first term is the Coulomb force, where $\rho_{e}$ is the free electric charge density, and **E** is the electric field. This force acts on the free charge in the electric field. The second and third terms represent the polarization forces acting on polar fluids, where ρ is the free electric charge density, and ε is the dielectric constant. In a homogeneous fluid at constant temperature (i.e., ▽·ε=0 ), the polarizing force is a gradient that can be described as: (18)f=▽·p
where *p* is the pressure. On the other hand, the electrostatic polarization force or dielectrophoretic force is: (19)$$f_{d}=P\cdot ▽E$$
where **P** is the polarization vector. The polarization force is only effective near the electrode, where the field gradient has the greatest effect. Moreover, the convective forces dominate in liquids with non-zero charge density. The main factor controlling the flow is the Coulomb force [23].

In the microscopic model, a fixed potential is assumed at the metal electrodes [12]. Therefore, the electrical boundary conditions on non-metallic substrates must be taken into account. When an electrolyte comes into contact with a surface, an electric double layer (EDL) forms in the absence of an external electric field [24,25]. Assuming ζ is the potential difference between the surface of the liquid and the electrically neutral body. By utilizing the Debye–Hückel approximation, the potential of the undisturbed EDL can be approximated as:(20)$$\phi_{EDL}≃\zeta e^{-\frac{z}{\lambda_{D}}}$$
where *z* is the distance to the interface, and the volumetric charge density within the Debye layer can be described as
(21)$$\rho_{EDL}=\;-\nabla \cdot \left(\varepsilon \nabla \phi_{EDL}\right)=\;-\frac{\varepsilon \zeta }{\lambda_{D}^{2}}\;e^{-\frac{z}{\lambda_{D}}}$$

Then, the total charge in the Debye layer can be expressed as: (22)$$Q_{EDL}=S\int_{0}^{\infty }\;\rho_{EDL}\left(z\right)dz=\;-\frac{\varepsilon \zeta S}{\lambda_{D}}$$
where *S* is the area of the interface. As the liquid is electrically neutral, the total charge in the Stern layer is $-Q_{EDL}$, resulting in a surface charge of
(23)$$\sigma_{S}=\frac{\varepsilon \zeta }{\lambda_{D}}$$

Equation (Equation 23) represents a surface charge on an infinite plane whose electric field is $\frac{\zeta }{\lambda_{D}}$. Then, the electric boundary conditions on the positive and negative electrodes are $\phi =\phi_{0}$ and ϕ=0, respectively. Moreover, the electric boundary conditions of the substrate are
(24)$$n\cdot \nabla \phi =\frac{\sigma_{S}}{\varepsilon }.$$

These equations can explain the mathematical model under appropriate boundary conditions. However, for irregular geometries where the voltage is a function of multiple coordinates, the mathematical model used to solve the described problem can become quite complex [19].

#### Pure Mathematical Model

Modeling EHD pump operation is a complex process governed by electrostatic and hydrodynamic partial differential equations. Manual solutions to these equations are complicated. Therefore, numerical methods are used to study and simulate EHD pump operation. The geometric parameters of the electrodes of the actuator and the working fluid can determine the performance of the EHD pump [7]. However, the modeling of parameters determined experimentally is costly. Therefore, the use of numerical simulation to predict the design of micro-pumps is used by many scholars. Experimentally validated simulation models are used primarily for fluidic EHD pump design, thermal management, and flow distribution. In addition to the mainstream finite element-based COMSOL Multiphysics commercial software, some open-source software such as OpenFOAM and Finite Element Method Magnetics can also be used for simulation. The software used in the numerical methods research in recent years are shown in Table 1.

Simulation software is mainly based on the finite element. Most fluidic EHD pump simulations have been performed with the commercial software COMSOL Multiphysics in recent years owing to its user-friendliness and reliable algorithms. The software can represent physical model equations and boundary conditions using the Electrostatics, Laminar Flow, and Transport of Diluted Species modules [40]. The solver allows the solution to obtain accurate results within a reasonable period.

The high cost of commercial software has encouraged some researchers to use open-source software, although COMSOL Multiphysics is used to conduct the majority of simulations. In addition, in future research, due to the recent rapid development of machine learning and increasing open-source projects, the authors believe that neural networks can be used to explore the relationship between various parameters of the working fluid and the operation performance of the EHD pump [46,47]. After the vigorous development of computing resources, more research and applications have been produced on the EHD pump.

## 4. Pump Structure and Material

This section introduces the EHD functional fluidic and gas pump’s base, electrode material, and geometry selection.

Currently, there are many EHD pump electrode geometries, such as needle, ring, planar, wire, mesh, and disk. These electrodes can form five basic electrode pairs in functional fluidic EHD pumps: thin planar electrode pairs, needle-to-ring electrode pairs, disk electrode pairs, triangular prism-to-silt electrodes (TPSE), and helical electrodes. Schematics of some electrode pairs are shown in Figure 4. Thin planar electrode pairs bear a low power density of 6.3 Wm/cm${}^{2}$[48] while needle-to-ring electrode pairs and TPSE are 111 Wm/cm${}^{2}$[49] and 17.5 Wm/cm${}^{2}$ [50]. Thin planar electrodes and TPSE generally use photomask patterning, which is easy to reproduce; however, needle ring electrodes are mainly assembled manually, with low reproducibility [51]. Moreover, thin planar electrodes and TPSE are fabricated with batches and are easy to integrate. However, needle ring electrodes are individually manufactured and hard to integrate.

Different electrode shapes can affect conduction-pumping properties such as static pressure, flow rate and efficiency [55]. The flow and pressure of an EHD pump increase as the applied voltage increases. The consumption of the pump is on the order of mW. The flow and pressure of the EHD functional fluidic pump depend on the pattern of the electrodes. Compared with needle ring and thin planar electrodes, triangular prism and silt electrode pair electrode patterns are widely used due to their high-output power density, and high compatibility with Micro-Electro-Mechanical System processes [39]. The electrode material can also affect the performance of the pump. For example, using parallel planar electrodes and HFE-7100 as the working fluid, the same design using copper electrodes requires more than twice the voltage to produce the same performance [56].

The base material of EHD pump can be hard or soft. Hard material includes silicon and glass [57]. Soft material includes PTFE, PET and PP. Rigid EHD pumps installed with rigid bases are used to drive the actuators, while soft EHD pumps loaded with flexible bases power the actuators and the soft parts of the bodies [58]. EHD pumps with flexible bases can achieve states such as bending and twisting.

The selection of base material, electrode material and geometry of functional fluidic EHD pumps in recent years is summarized in Table 2.

Electrode materials such as aluminum, copper, titanium, and steel are commonly corroded by corona [69,70]. Noll et al. performed ionization tests using germanium and silicon needles to overcome corrosion effects [71]. Sometimes, it is also possible to apply a corrosion-resistant material to the electrode surface [72].

EHD gas pumps mainly use electrode pairs such as wire-(rod/ring/plate/mesh), mesh-(rod/ring), and needle-(plate/ring). Figure 5 shows several representative electrode pairs. Moreover, most studies choose acrylic and glass as the material for the discharge chamber.

Electrode polarity, number of electrodes, and orientation of electrodes affect the flow of working gas in an EHD pump. Zhang and Lai [76] tried configuring 28 emitter electrodes in a square channel. After increasing the number of emitter electrodes, the electric field tends to become more uniform; however, there is also a limit to the volume flow that an EHD gas pump can produce. After adding the secondary emitter electrode, as reported by Chang et al. [77], the secondary electrode can effectively increase the volume flow generated by the single primary electrode. However, its performance depends on the number of secondary electrodes added and the gap distance between primary and secondary electrodes. Another way to increase the volume flow rate is to use new electrode configurations, such as needle array-to-mesh [78,79] and wire array-to-mesh [27]. While these new electrode configurations generally result in higher volumetric flow rates, maintenance may become burdensome with application. Moreover, multi-stage EHD pumps [80,81] are more efficient than simply increasing the applied voltage of a single-stage pump if the flow distance is expected to be extended [82]. Investigations connecting multiple corona wind turbines in series to extend the EHD-induced flow over longer distances, confirmed that additional stages could increase the volumetric flow produced by a single individual stage [83].

The selection of base material, electrode material, and geometry of EHD gas pumps in recent years’ research are summarized in Table 3.

## 5. Working Medium

To operate EHD pumps, the working medium must be dielectric fluid. Furthermore, free charges must be present in the fluid to establish an electric field. In a corona discharge, free charges can be generated in the air by breaking down neutral air molecules in the ionization zone near the emitter electrode. With the application of a high electric field, the generated ions or electrons are repelled by the emitter electrode and migrate towards the collector electrode by Coulomb force. During transport, collisions between charged particles and neutral air molecules create pairs of ions or electrons.

To select a suitable working medium for an EHD pump, several factors should be considered:Conductivity: The working medium must be a dielectric material, otherwise the EHD phenomenon will not occur [7,91]. For example, to achieve a stable pulsed-jet mode of ejection in [92], the electrical conductivity of inks should not exceed 10${}^{-5}$ S/m.Viscosity: The working medium should have a low viscosity in order to minimize energy loss due to friction [7,93].Corrosion: The working medium should be compatible with the materials of the pump, as certain fluids can cause corrosion or other damage to the pump components [94,95].Temperature: The working medium should be able to withstand the temperature range of the application, and not cause any thermal changes or decomposition in the fluid [96].

### 5.1. Functional Fluidic EHD Pump

The working fluid is confined between the micropatterned electrodes. When the electrodes are charged, conduction occurs due to the positive and negative ions produced by the dissociation of liquid molecules in the presence of a high electric field. The charged surface attracts counterions and repels co-ions (on the dielectric liquid side), forming a heterogeneous charge layer away from the double layer. When an electric field with a component of the electrode plates is applied, Coulomb forces will be exerted on these mobile ions, and the electromigration of the heterogeneous charge layer drags the bulk fluid through viscous interactions. The working medium of functional fluidic EHD pump used in recent years are summarized in Table 4. HFE-7100, HFE-7200, HFE-7300, and dibutyl decanedioate (DBD) were popularly used as the working fluid in many studies. Their specific parameters are shown in Table 5 [55].

Impurities on the electrode surface strongly influence the current flowing through the dielectric liquid because the impurities appear as ions. In addition, the pressure generated by the Coulomb force is greatly affected by impurities. Therefore, in order to remove impurities, the electrodes need to be cleaned.

### 5.2. EHD Gas Pump

Different from functional fluidic EHD pumps, the working medium in EHD gas pumps is gas.

The electrical characteristics of surface discharges are compassionate as a function of different atmospheric parameters such as ambient gas humidity, pressure temperature [104]. More details can be found in [105]. To summarize, some trends are given as follows:When the relative air humidity increases, the discharge stability decreases [106];Pressure plays a fundamental role on discharge below 60 °C while temperature has no effect [107];The surface of the dielectric wall will greatly affect the electrical characteristics of the discharge [108];Airflow alters the physical mechanism if discharged in free airflow and significantly limits the transition from glow to arc [109].

HFE-7100, HFE-7200, HFE-7300 and dibutyl decanedioate (DBD) have been widely used as the working gas in many studies. Their specific parameters are shown in Table 6. In addition, the working medium in EHD gas pumps used in recent years is summarized in Table 7.

## 6. Recent Progress

The top 10 journals that currently accept studies on the research topic of EHD pumps are depicted in Figure 6 [131]. The data reveal that the majority of the research results are disseminated in journals within the fields of electrical science, mechanical science, and thermodynamics.

Due to recent developments in microelectronics, functional fluidic EHD pumps have attracted great attention from many researchers in the field of soft actuator technology. Additionally, EHD air pumps are considered a viable replacement for conventional fans or pumps in some applications, and can be used for electronics cooling, microfluidics device and biology. Moreover, EHD air pumps also have applications in flow control and drying enhancement [132,133,134,135]; however, this paper does not introduce them in detail. Furthermore, with the recent successful flight demonstration of an EHD propulsion aircraft by Xu et al. [136], there has been increasing excitement and enthusiasm around the study of EHD technology.

### 6.1. Soft Robotics

Functional fluidic EHD pumps have been widely used in driving actuators. The mechanism uses a high electrical voltage to induce dynamic motion in a fluid medium, converting electrical energy into mechanical output. EHD pumps are generally quieter and smaller than high-pressure gas-driven soft robots such as [137,138,139]. For present applications in soft pumps, EHD pumps using the ion-drag working mode represent the primary type owing to high-flow rates and relatively simple design [140,141]. In 2019, a popular soft fluid EHD pump was the soft stretchable pump proposed by Cacucciolo et al. [58]. Cacucciolo et al. have designed and developed four generations of stretchable pumps. The first generation circuit consisted of five tilting capacitors spaced 1 mm apart. It operates on the EHD conduction pump principle. The second generation is a scaled-down version of the first generation, with half the channel size, half the electrode gap, and 43 tilting capacitors. The third generation relies on a charge injection mechanism and thus exhibits a different electrode layout: two series of comb planar electrodes face each other. The fourth generation has the same geometry as the third generation. However, the electrode material has been replaced by commercial silver ink because using silver particles as the conductive material can solve the performance degradation of carbon-based electrodes. In this structure, two pairs of comb planar electrodes are responsible for ionizing the liquid molecules and moving the ionized molecules from left to right.

Recently, many soft actuators have been inspired by this design. A soft fiber-reinforced bending finger (Figure 7a) was fabricated by Mao et al., and one chamber of the soft finger expands with the internal pressure generated by the pump, while the other two chambers, which act as containers for the working medium, contract, eventually causing the finger to bend [52]. Subsequently, this finger-type actuator realized bidirectional motion by Nagaoka et al. [63] (Figure 7b). Moreover, Mao et al. (Figure 7c) proposed an eccentric actuator, which can bend as the applied voltage increases [142]. Recently, a novel rolling robot (Figure 7d) was designed consisting of a flexible EHD pump and a layer of natural latex, allowing the robot to roll as the working dielectric oscillates using a high-voltage circuit [68]. Apart from this, Thongking et al. implemented object grabbing using a soft robotic gripper [143] and Mao et al. designed a droplet sorter driven by a functional fluidic pump [144]. Functional fluidic pumps provide the basis for many soft robot designs because they are soft and work quietly; however, their performance is insufficient for many high-power applications.

### 6.2. Cooling Device

Many researchers, such as Yabe et al. [145], Singh et al. [146], Bryan and Seyed-Yagoobi [147], and Cotton [148] have studied the effect of DC electric fields on thermal convection. They found that the heat transfer enhancement and pressure drop penalty increased with the increasing applied voltage, and the maximum is limited by the breakdown voltage. Depending on the geometry and electrode configuration, the voltage drop loss due to EHD can fluctuate widely (between 2 and 15 times). Moreover, Feng et al. described the effective management of liquid-flow distribution between two branch lines using an EHD conduction pump at total mass flux rates of 100 and 200 kg/m${}^{2}$ s [149]. Furthermore, Vasilkov et al. discovered that Onsager’s theory can be applied to field enhancement by simulating the enhancement of heat transfer and dissociation in relation to the EHD process [150]. Heat transfer enhancements in the range of 200–300% are typical [118].

EHD cooling devices are designed in various shapes. O’Connor et al. used a comb and circular-shaped electrodes (Figure 8a), allowing the pump to produce significant flow rates with and without bending [151]. Zeng et al. used EHD gas pumps with needle electrode strips (Figure 8b) to investigate the effects of the needle-electrode arrangement, the structure of the mesh electrodes, the gap between the needle tip and the mesh, and the distance on the output speed and power consumption of the ion air pump [152]. The optimized ion air pump can reduce the surface temperature of the 200 nW power heat source by more than 30 °C. Tien et al. tested effectiveness in maintaining and increasing volumetric flow and reducing power requirements using multiple single-stage EHD air pumps (Figure 8c) [153]. Ramadhan et al. proposed a new cooling system design that can be used in consumer electronic products, such as notebook computers, using EHD pumps (Figure 8d) [78].

In addition, multi-phase flow represents an interest of many researchers. For example, Pearson and Seyed-Yagoobi created a configuration of pumping in a radial direction, which minimized the distance the working fluid had to travel [154], and Patel et al. were successful in utilizing EHD pumps for electrowetting of heaters during film flow boiling under microgravity conditions [155].

EHD gas pumps can be used for thermal management because of the potential to remove heat from high-density flux heat sources [43] and no mechanical parts are involved during operation. In addition, it is essential to determine the inlet design parameters of the EHD pump, such as the nature of the working medium, and the pump size, which directly affect heat transfer performance. For example, Blanc et al. [156] experimented with functional fluidic EHD pumps for heat pipe solutions for the thermal management of embedded missiles. O’Connor et al. focused on electrical equipment cooling and proposed the possibility of being incorporated into flexible electronics [151]. Moreover, Nishikawara and Yagoobi presented the pumping performance of a micropump with three pairs of perforated electrodes designed to be fabricated and integrated into a high-heat flux evaporator embedded in an electronics cooling system [157]. Recent studies have demonstrated the possibility of functional EHD pumps to provide cooling support for applications in military, aerospace, and consumer electronics.

### 6.3. Future Research

In the realm of EHD pumps, there exist a number of limitations that must be taken into account during the design and implementation process. These include: the small-scale nature of the pumps, the tendency to operate optimally with low-viscosity fluids [93] and low flow rates, and the requirement for dielectric fluids and susceptibility to corrosion in harsh environments [69,95]. Despite these limitations, ongoing research efforts aim to address and overcome these challenges, ultimately leading to improved design and performance of EHD pumps.

There are numerous research opportunities that could enhance the capabilities and applications of EHD pumps in the future. These include the creation of microscale EHD pumps, studying multiphase flow, improving efficiency, high-temperature operation, biomedical applications, and mathematical modeling.

The miniaturization of EHD pumps could open new avenues for use in small-scale devices and microfluidic systems [158]. Research into multiphase fluid pumping could broaden the range of materials that can be pumped with EHD pumps [159]. Efforts to improve efficiency through new electrode configurations and materials could reduce energy consumption and operational costs. High-temperature EHD pumping research could open up new possibilities in industries such as power generation and aerospace [130,160]. Biomedical applications, such as drug delivery [161] and cell separation [162], could be possible with EHD pumps.

In addition, mathematical modeling and simulation can help optimize the performance and efficiency of EHD pumps. Machine learning can be a powerful tool in EHD pump research for modeling, optimization and control. Through machine learning algorithms, researchers can create more precise mathematical models of EHD pumps and optimize performance by determining the optimal operating conditions and the best electrode configurations and materials [47]. Machine learning can also be used to identify and diagnose problems with the pumps and schedule maintenance accordingly [46]. Additionally, machine learning can assist in the development of control strategies, such as adaptive control, that can enhance the performance of EHD pumps [163]. By integrating machine learning with computational tools in EHD pump research, more efficient and reliable EHD pumps can be developed for various industrial applications.

## 7. Conclusions

This article reviews the latest developments in functional fluidic and gas EHD pumps and summarizes the selection of pump structures, electrodes, and working media in recent studies. EHD pumps are rapidly developing and bear significant research potential in multiple disciplines. Industrially, EHD pumps have practical applications in various industries, such as robotics, aerospace, and electronics, and demonstrate developmental potential in the consumer market. The recent literature on practical devices, based on EHD pumps, has focused on soft robotics design, electronics cooling and microfluidics. The authors are optimistic about the future of this technology in soft robotics applications.

## Figures and Tables

**Figure 1 micromachines-14-00321-f001:**
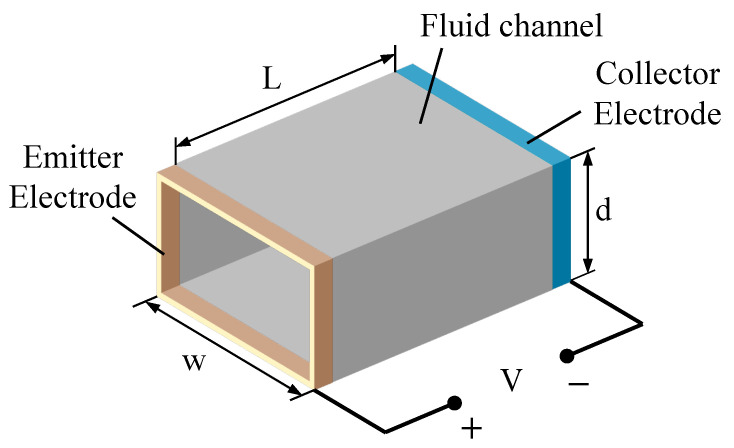
Fluid EHD pump geometry [7]. *L*, *d* and *w* represent channel length, depth and width, respectively. *V* denotes the high-voltage power source.

**Figure 2 micromachines-14-00321-f002:**
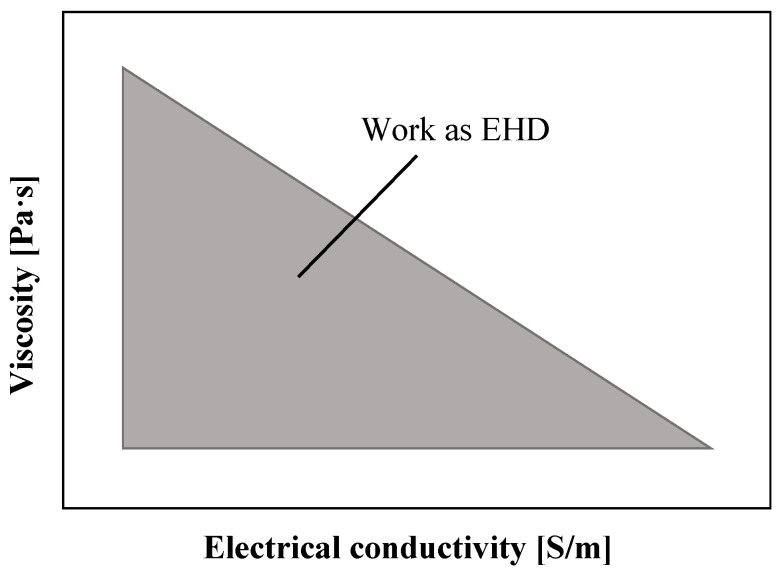
Distribution of working fluids in the electro-conjugate fluid triangle (log scale) [17]. The area inside the gray triangle is available for working fluids.

**Figure 3 micromachines-14-00321-f003:**
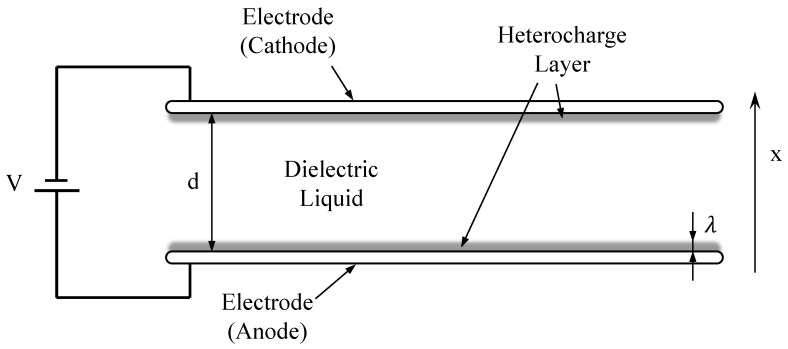
Parallel electrodes in dielectric liquid. V is the direct current (DC) supply, d is the electrode spacing, λ is the thickness of the heterocharge layer, and x is characteristic length for the normalization of the variable.

**Figure 4 micromachines-14-00321-f004:**
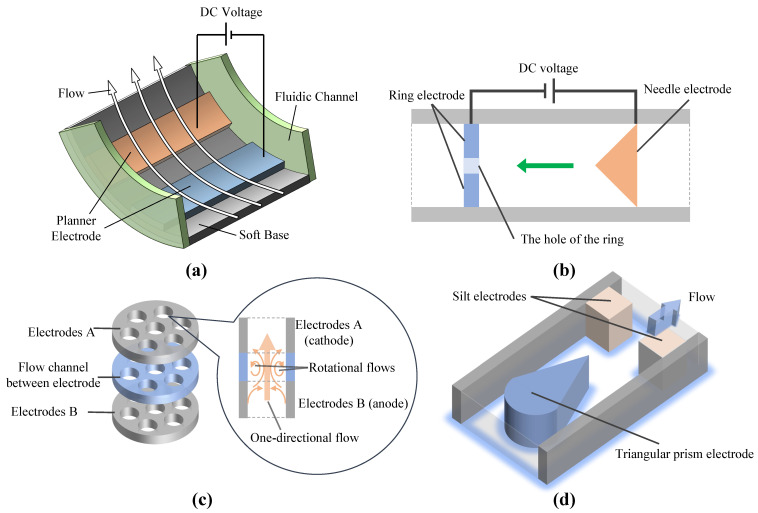
Schematic of functional fluidic EHD pump with various electrode types. (**a**) Planar electrodes [38]. (**b**) Needle-to-ring electrodes [52]. (**c**) Disk electrodes [53]. (**d**) Triangular prism-to-silt electrodes [17,39,54].

**Figure 5 micromachines-14-00321-f005:**
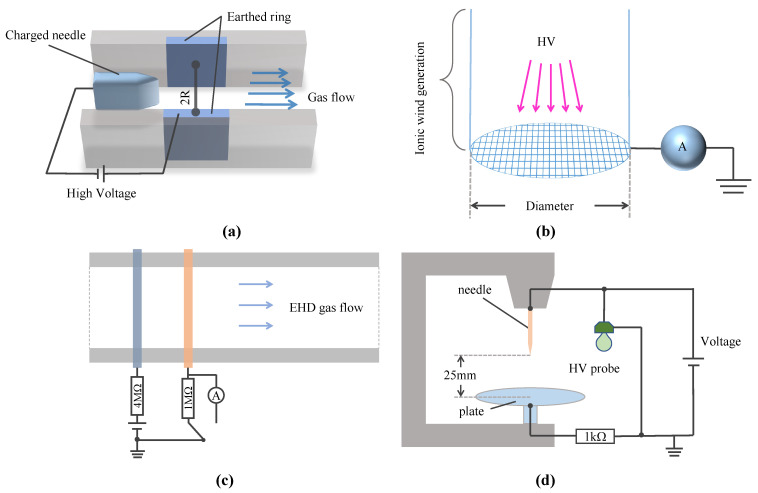
Schematic of EHD gas pump with various electrode types. (**a**) Needle-ring electrodes [73]. (**b**) Mesh-ring electrodes [4]. (**c**) Wire-rod electrodes [74]. (**d**) Needle-plate electrodes [75].

**Figure 6 micromachines-14-00321-f006:**
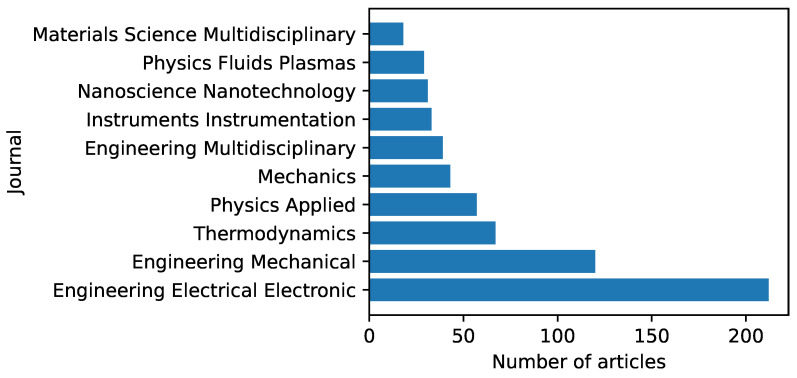
Distribution of accepted journals.

**Figure 7 micromachines-14-00321-f007:**
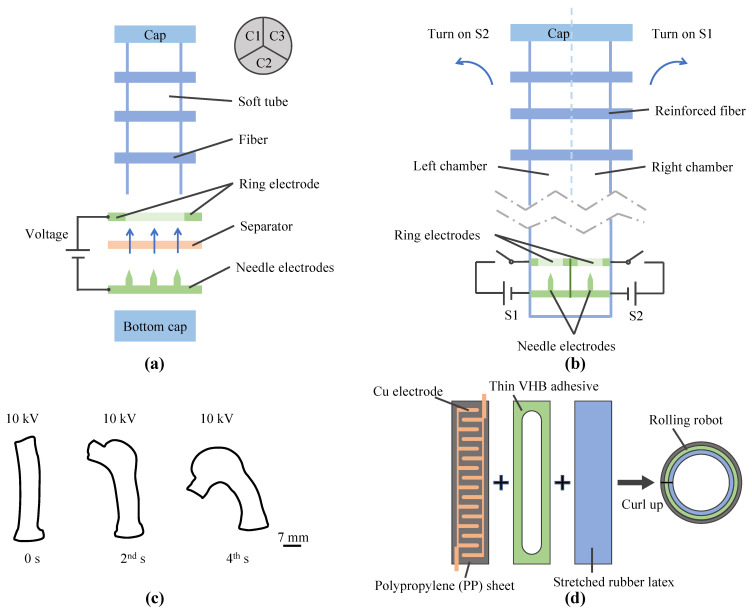
Soft robots driven by EHD pumps. (**a**) Soft fiber-reinforced bending finger [52]. (**b**) Soft finger with bidirectional motion [63]. (**c**) Eccentric actuator [142]. (**d**) Fluidic rolling robot [68].

**Figure 8 micromachines-14-00321-f008:**
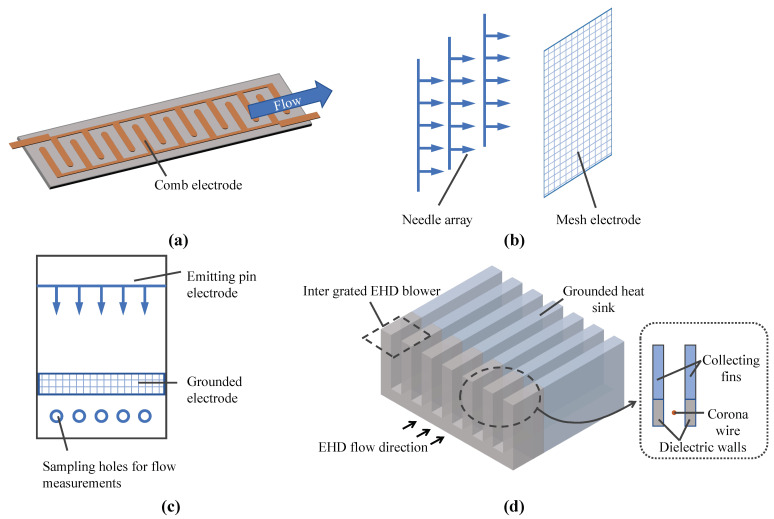
Cooling device. (**a**) Pump produces significant flow rates with and without bending [151] (**b**) Air pump for multi-parameter testing [152] (**c**) Multiple single-stage EHD air pumps [153] (**d**) Notebook computer cooling system [78].

**Table 1 micromachines-14-00321-t001:** Numerical methodology for fluidic EHD pump simulations in recent years.

Year	Author	Problem Type	Software
2008	Jewell et al. [26]	Flow analysis	COMSOL Multiphysics
2014	Fylladitakis et al. [27]	EHD pumps design	FEMM
2014	Ongkodjojo et al. [28]	Thermal management	COMSOL Multiphysics
2015	Chirkov et al. [29]	Flow analysis	COMSOL Multiphysics
2016	Luo et al. [30]	Flow analysis	-
2016	Yang et al. [31]	Flow analysis	COMSOL Multiphysics
2017	Kuwajima et al. [32]	EHD pumps design	COMSOL Multiphysics
2017	Sato et al. [33]	EHD pumps design	COMSOL Multiphysics
2018	Ramadhan et al. [34]	Thermal management	COMSOL Multiphysics
2020	Adamiak [35]	Flow analysis	COMSOL Multiphysics and ANSYS
2020	O’Connor and Seyed-Yagoobi [36]	Flow analysis	COMSOL Multiphysics
2020	Nourdanesh et al. [37]	Flow analysis	COMSOL Multiphysics
2020, 2021	Mao et al. [38,39]	Flow analysis	COMSOL Multiphysics
2021	Talmor and Seyed-Yagoobi [40]	Flow analysis	COMSOL Multiphysics
2021	Wang et al. [41]	Flow analysis	(GPU-based computation)
2021	Monayem [42]	Flow analysis	ANSYS/FLUENT
2022	Selvakumar et al. [43]	Thermal management	OpenFOAM
2022	Mazumder et al.	Thermal management [44], flow analysis [45]	ANSYS/FLUENT

**Table 2 micromachines-14-00321-t002:** Summarized material and geometry of functional fluidic EHD pumps in recent years.

Year	Author	Electrode Material	Base Material	Electrode Geometry
2010	Yu et al. [56]	Copper, silver and gold	Silicon	Planar
2011	Seyed-Yagoobi et al. [59,60]	Stainless steel	PTFE	Disk
2013	Kano and Nishina [55]	Cr/Au	Glass	Planar
2014	Patel and Seyed-Yagoobi [61]	Stainless steel	PTFE	Disk
2016	Russel et al. [62]	Au	Glass	Planar
2019	Nagaoka et al. [63]	Tungsten and brass	Glass	TPSEs and needle-to-ring
2019	Tsukiji et al. [53]	Copper	-	Disk
2019, 2020	Mao et al. [17,39,54]	Ni/Au	Glass	TPSEs
2020	Seki et al. [64]	Copper	PET	Planar
2020	Mao et al. [52]	Tungsten and brass	Glass	Needle-to-ring
2020	Vasilkov et al. [65]	Aluminum	PET	Ring
2020	O’Connor and Seyed-Yagoobi [36]	-	-	Helical
2020	Blanc et al. [66]	Copper	PTFE	Disk
2020	Murakami et al. [67]	Copper	PP	Planar
2022	Mao et al. [68]	Copper	PP	Planar

**Table 3 micromachines-14-00321-t003:** Summarized material and geometry of EHD gas pumps in recent years.

Year	Author	Electrode Material	Discharge Chamber Material	Electrode Geometry	Interelectrode Gap [mm]
2006	Rickard et al. [73]	Copper and steel	Acrylic	Needle-ring	12.5
2008	Moreau and Touchard [4]	-	Plexiglas	Mesh-ring	5, 10, 15, 20
2011	Takeuchi et al. [74]	-	Acrylic	Wire-rod	13
2015	Birhane et al. [82]	Copper	Acrylic	Ring-wire	20, 50
2015	Quan et al. [84]	Stainless-steel	Glass	Mesh-rod	-
2016	Mizeraczyk et al. [85]	Stainless-steel	Acrylic	Needle-plate	25
2018	Moreau et al. [75]	Stainless-steel	-	Needle-plate	25
2019	Gao et al. [86]	Stainless-steel	Glass	Mesh-rod	-
2020	Qu et al. [87]	Tungsten and stainless-steel	Acrylic	Needle-ring	6, 11, 16, 21, 26
2020	Chang et al. [77]	Copper	Acrylic	Ring	15, 20
2022	Rubinetti et al. [88]	Stainless-steel	Acrylic glass	Mesh-wire	50
2022	Yang et al. [89]	Stainless-steel	Acrylic glass	Mesh-wire	10
2022	Li et al. [90]	Tungsten, copper	Acrylic	Plate-wire	5

**Table 4 micromachines-14-00321-t004:** Working medium adoption of functional fluidic EHD pumps in recent years.

Year	Author	Working Medium
2007	Lee et al. [97]	HFE-7100
2010	Yu et al. [56]	HFE-7100
2011	Seyed-Yagoobi et al. [59,60]	HCFC-123
2013	Kano and Nishina [55]	HFE-7100
2014	Patel and Seyed-Yagoobi [61]	HCFC-123
2016	Russel et al. [62]	HFE-7100
2019	Nagaoka et al. [63]	FF-1EHA2
2018	Tsukiji et al. [98]	HFE-7100
2019	Tsukiji et al. [53]	HFE-7300
2019	Mao et al. [17]	DBD and HFE-7200
2019, 2020	Mao et al. [39,54]	DBD
2020	Seki et al. [64]	HFE-7300
2020	Mao et al. [52]	FF-1EHA2 and FF-101EHA2
2020	Vasilkov et al. [65]	GK-1700 Transformer oil
2020	O’Connor and Seyed-Yagoobi [36]	HFE-7100
2020	Blanc et al. [66]	HFE-7000, 7100, 7200, 7300, 7500 and Acetone
2020	Murakami et al. [67]	HFE-7300
2022	Mao et al. [68]	HFE-7300

**Table 5 micromachines-14-00321-t005:** HFE-7100 and HFE-7300 properties.

Property	HFE-7100 [99,100]	HFE-7200 [101,102]	HFE-7300 [99,103]	DBD [17]
Chemical formula	$C_{5}H_{3}F_{9}O$	$C_{2}F_{9}OC_{2}H_{5}$	$C_{7}H_{3}F_{13}O$	$C_{18}H_{34}O_{4}$
Kinematic Viscosity [cSt]	0.38	0.61	0.71	-
Dielectric constant @ 1 kHz	7.4	7.3	6.1	4.54
Liquid mass density [g/cm${}^{3}$]	1.51	1.43	1.67	0.94
Dielectric Strength Range, 0.1″ gap [kV]	>25	>25	>25	-
Electrical conductivity [S/m]	-	$67\times 10^{-9}$	-	$5.6\times 10^{-9}$

**Table 6 micromachines-14-00321-t006:** Some dielectric gas properties (Atmospheric pressure).

Gas	Formula	Breakdown Voltages (300 K) [kV/cm]	Density (273.15 K) [g/L]
Air [110,111]	Mixture	30	1.2
Argon [112,113]	Ar	20	1.78
Nitrogen [112,114]	$N_{2}$	40.1	1.25
R134a [115,116]	$C_{2}H_{2}F_{4}$	31	-

**Table 7 micromachines-14-00321-t007:** Working medium adoption of EHD gas pumps in recent years.

Year	Author	Working Medium
2003	Feng and Seyed-Yagoobi [117]	R134a
2011	Takeuchi et al. [74]	Air
2014	Nangle-Smith and Cotton [118]	R134a
2015	Dinani et al. [119]	Air
2015	Birhane et al. [82]	Air
2016	Mizeraczyk et al. [85]	Air
2018	Van Doremaele et al. [120]	Helium
2018	Moreau et al. [75]	Air
2018	Granados and Victor [121]	Argon, Nitrogen, Oxygen
2018	Nangle et al. [122]	R134a
2019	Reiser et al. [123]	Argon
2019	Gao et al. [86]	R113
2020	Chi et al. [124]	Nitrogen
2020	Calvo et al. [125]	Argon
2020	Polat and Izli [126]	Air
2022	Calvo et al. [127]	Argon, Xenon
2022	Dima et al. [128]	Nitrogen
2022	Hassan and Mohammad [129]	Nitrogen
2022	Li et al. [90]	Air
2022	Singh et al. [130]	R134a

## Data Availability

Not applicable.

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
