# Peer review of "A Review on Electrohydrodynamic (EHD) Pump"

_micromachines, 2023, doi:10.3390/mi14020321_

Round 1

Author Response

Dear reviewer,

Thanks for your kind suggestions.

Please find the enclosed Response to reviewer.

Thanks for your precious time and have a good day.

Best regards

Reviewer 2 Report

The authors undertook a "Review on Electrohydrodynamic (EHD) Pump". While I find that this review is useful, the authors need to do a more through and critical literature review of existing studies rather than merely reporting them.

The authors could consider elaborating on limitations of current EHD pumps in different applications, clearly identify knowledge gaps that currently exists, and which in turn could help researchers investigate future potential solutions.

1. What is the main question addressed by the research? In this manuscript, the authors undertook a detailed literature review of EHD pumps including the applications in different sectors including healthcare and electronics.   2. Do you consider the topic original or relevant in the field? Does it address a specific gap in the field? While the topic under investigation is relevant, the authors did not conduct a critical review of the state-of-the-art. They merely reviewed current studies without robustly establishing knowledge gaps how this could potentially be addressed.
3. What does it add to the subject area compared with other published material? None. It merely just collates findings from previous work
4. What specific improvements should the authors consider regarding the methodology? What further controls should be considered? The authors should consider critically reviewing previous studies by highlighting limitations, applications, and how future studies could help fill these knowledge gaps. They should specifically point out potential future research directions and tools that can help facilitate these studies. Could Artificial Intelligence, Machine Learning, Computational Tools add to the discourse? What are the current limitations of these tools and can these be addressed?
5. Are the conclusions consistent with the evidence and arguments presented and do they address the main question posed? While the conclusion is consistent, more robust evidence and arguments are necessary. Listing some quantifiable highlights of the study is also useful. For example, ranges of applicability of certain EHD pumps.
6. Are the references appropriate? Yes.
7. Please include any additional comments on the tables and figures. One of the main contributions of review articles is to critically review existing studies and propose future research directions. While the current manuscript have done this in part, proposing future research directions should be included and a bit more critique of current studies' limitations are necessary.

Author Response

(The authors gave the same response as above.)

Reviewer 3 Report

The authors a Review paper on the Electrohydrodynamic Pumps.

The paper is interesting and generally well prepared

The abstract is to be extended.

The novelty of the paper is to be clearly stated.

A bibliometric study (yearly published papers related to the subject), based on Scopus or Web of science database is to be performed.

A table summarizing the most important techniques and configurations of EHD pumps and including the advantages and disadvantages is to be added.

References are to be added to Fig. 8

The discussion is to be improved by considering economic and environmental aspects.

A nomenclature is to be added.

Author Response

(The authors gave the same response as above.)

Round 2

Reviewer 2 Report

The authors have done a good job in significantly improving the manuscript and implementing reviewers' comments.